ACPT gene is inactivated in mammalian lineages that lack enamel or teeth

Mu Yuan
Huang Xin
Liu Rui
Gai Yulin
Liang Na
Yin Daiqing
Shan Lei
Xu Shixia
Yang Guang gyang@njnu.edu.cn
Jiangsu Key Laboratory for Biodiversity and Biotechnology, College of Life Sciences, Nanjing Normal University , Nanjing, Jiangsu , China
Gillespie Joseph
Electronic publication date: 2021 Jan 22
Publication date: 2021
Volume: 9
Electronic Location ID: e10219
Received 2020 Apr 6; Accepted 2020 Sep 29
Copyright: © 2021 Mu et al.
Copyright year: 2021
Copyright holder: Mu et al.
License: This is an open access article distributed under the terms of the Creative Commons Attribution License, which permits unrestricted use, distribution, reproduction and adaptation in any medium and for any purpose provided that it is properly attributed. For attribution, the original author(s), title, publication source (PeerJ) and either DOI or URL of the article must be cited.
License URL: https://creativecommons.org/licenses/by/4.0/

Keywords: ACPT, Tooth evolution, Enamel loss, Mammals, Pseudogene, Inactivation time

Funding: NSFC 31630071 Ministry of Science and Technology of China 2016YFC0503200 NSFC 31570379 and 31772448 Jiangsu Higher Education Institutions This work was supported by the Key Project of the NSFC (grant no. 31630071), the National Key Program of Research and Development, Ministry of Science and Technology of China (No. 2016YFC0503200), the NSFC (grant no. 31570379, 31772448), and the Priority Academic Program Development of Jiangsu Higher Education Institutions (PAPD). The funders had a role in study design, data collection and analysis, decision to publish, and preparation of the manuscript.

==============================
Loss of tooth or enamel is widespread in multiple mammal lineages. Although several studies have been reported, the evolutionary mechanisms of tooth/enamel loss are still unclear. Most previous studies have found that some tooth-related genes have been inactivated in toothless and/or enamel-less mammals, such as ENAM, ODAM, C4orf26, AMBN, AMTN, DSPP, etc. Here, we conducted evolutionary analyses on ACPT playing a key role in amelogenesis, to interrogate the mechanisms. We obtained the ACPT sequences from 116 species, including edentulous and enamel-less mammals. The results shows that variant ORF-disrupting mutations were detected in ACPT coding region among nine edentulous baleen whales and three enamel-less taxa (pygmy sperm whale, aardvark, nine-banded armadillo). Furtherly, selective pressure uncovered that the selective constraints have been relaxed among all toothless and enamel-less lineages. Moreover, our results support the hypothesis that mineralized teeth were lost or degenerated in the common ancestor of crown Mysticeti through two shared single-base sites deletion in exon 4 and 5 of ACPT among all living baleen whales. DN/dS values on transitional branches were used to estimate ACPT inactivation records. In the case of aardvark, inactivation of ACPT was estimated at ~23.60–28.32 Ma, which is earlier than oldest aardvark fossil record (Orycteropus minutus, ~19 Ma), suggesting that ACPT inactivation may result in degeneration or loss of enamel. Conversely, the inactivation time of ACPT estimated in armadillo (~10.18–11.30 Ma) is later than oldest fossil record, suggesting that inactivation of ACPT may result from degeneration or loss of enamel in these mammals. Our findings suggested that different mechanisms of degeneration of tooth/enamel might exist among toothless and enamel-less lineages during evolution. Our study further considered that ACPT is a novel gene for studying tooth evolution.

Introduction

Dental innovations (such as differentiated dentitions and the evolution of tri-bosphenic molar) have been regarded as the great success of mammalian evolution and adaptation (Ungar, 2010). However, in spite of their importance for animal survival, teeth have been lost independently in multiple mammalian lineages, such as baleen whales and pangolins. In addition, some lineages have lost their outer enamel of teeth, such as pygmy sperm whale and dwarf sperm whale, aardvarks and species from Xenarthra (Davit-Béal, Tucker & Sire, 2009). Tooth loss and/or enamel loss is one of the most important field for mammalian tooth evolution.

Amelogenesis imperfecta (AI) and tooth loss are the diseases that characterized by genetic defects in the formation of enamel and teeth. Multiple studies have suggested these genetic disorders are mainly caused by mutations of protein-coding genes functioned in formation of enamel and teeth (Stephanopoulos, Garefalaki & Lyroudia, 2005; Smith et al., 2017b). Of these genes, three enamel matrix protein genes (EMPs, i.e., AMELX, AMBN and ENAM), two enamel proteases genes (MMP20 and KLK4), and some other related genes (e.g., C4orf26, AMTN, ODAM, ACPT, DSPP) have been confirmed to be candidate genes responsible for the diseases (Crawford, Aldred & Bloch-Zupan, 2007; Smith et al., 2017b). The variant inactivating mutations have been detected in these genes among toothless and enamel-less mammalian lineages. However, the mechanisms of tooth loss or enamel loss are still completely unclear.

It has been reported that ACPT was lower expressed in testicular cancer tissues compared to normal tissues and is regulated by steroid hormones (Yousef et al., 2001). Besides, ACPT is also expressed in the brain and acts as a tyrosine phosphatase to modulate signals mediated by ErbB4 (Fleisig, El-Husseini & Vincent, 2004). But, it is interesting to note that ACPT is expressed in secretory-stage ameloblasts (Seymen et al., 2016), which can induce odontoblasts differentiation, mineralization of dentin, and amelogenesis (Choi et al., 2016). Furthermore, there are some increasing evidences that homozygous missense variants of ACPT would lead to AI (e.g., c.226C>T, p.Arg76Cys; c.746C4T, p.P249L) (Seymen et al., 2016; Smith et al., 2017a). These evidences suggested that ACPT play an important role in amelogenesis.

All extant Mysticeti, descended from toothed ancestors, have no teeth and instead have baleen (Uhen, 2010). Paleontological evidences have shown that mineralized teeth were lost in the common ancestor of crown Mysticeti. Moreover, a transitional stage from tooth to baleen in stem mysticetes have been revealed in some taxa bearing both teeth and baleen (Deméré et al., 2008). Although many tooth-related genes have been revealed to be inactivated in various living mysticetes (e.g., AMBN, ENAM, AMEL, AMTN, MMP20, C4orf26 and DSPP) (Deméré et al., 2008; Meredith et al., 2009, 2011; Gasse, Silvent & Sire, 2012; Delsuc, Gasse & Sire, 2015; Springer et al., 2016, 2019), only the MMP20 are commonly inactivated across all the living baleen whales (Meredith et al., 2011). This molecular evidence is consistent with earlier studies of paleontology and anatomy.

Despite its significance in mammalian enamel maturation, very little is known about ACPT evolutionary trajectory, relationship and function in mammals. To address this issue, we carried out a series of evolutionary analyses on ACPT, aim to uncover the evolutionary pattern of ACPT gene among mammalian lineages.

Methods

Sequences mining and BLAST searches

The full-length coding sequences (CDS) of ACPT gene were extracted from the OrthoMaM v10b (http://orthomam1.mbb.univ-montp2.fr:8080/OrthoMaM_v10b5/), ENSEMBL (http://www.ensembl.org/index.html?redirect=no) and NCBI (https://www.ncbi.nlm.nih.gov/) databases (Table S1). ACPT of some whales were extracted from their Genome and SRA database of NCBI (Tables S2 and S3). To further ensure the sites of inactivating mutation of toothless/enamel-less lineages, we used the CDSs of some representative placental species with well-annotated genomes (Homo sapiens (human), Canis lupus familiaris (Dog), Bos taurus (Cow), Echinops telfairi (Lesser hedgehog tenrec)) as queries including ∼50 bp of flanking sequence on each exon. These sequences were used as queries to BLAST against toothless/enamel-less mammals to confirm the related inactivating mutation among baleen whales.

Identification of inactivating mutations and functional sites and domains

The intact ACPT sequences (human, cow, tenrec) were used for identifying inactivating mutations (including mutation of initiation codons, frame-shift insertions and deletions, premature stop codons, splice sites mutation of intron/exon boundary (GT/AG), etc.). The inactivating mutation was identified based on BLAST searches against whole genomes of the relevant taxon from NCBI. The information on gene function, related key amino acid sites/domains was searched from UniProtKB/Swiss-Prot (http://www.uniprot.org/) and some references.

Alignment and phylogenetic analysis of mammalian ACPT

The 116 mammalian ACPT sequences were aligned based on their amino acid translations using online PRANK (https://www.ebi.ac.uk/goldman-srv/webprank/), and then deleted the gaps and non-homologous regions by using GBLOCK, then we corrected the multiple sequences alignment (MSA) in MEGA 7 (Kumar, Stecher & Tamura, 2016) by eye.

A gene tree was reconstructed by Mrbayes 3.2 (Ronquist et al., 2012) with a general time reversible (GTR) substitution model and rate heterogeneity modeled with a Gamma distribution, as conducted by MrModeltest version 2 using the Akaike information criterion (AIC) (Nylander, 2004). In bayesian analysis, four simultaneous runs with four chains each were run for two million generations, sampling every 1,000 trees. The first 25% of these trees were discarded as burn-in when computing the consensus tree. Tracer v1.5 software was used for checking convergence among chains in Bayesian analysis. When the ESS value is higher than 200, and the average standard deviation of spilt frequencies is lower than 0.01, we think it reach convergence level.

Selection analyses

To evaluate the selective pressure of relevant branches leading to enamel-less and toothless lineages respectively, we implemented two ratio branch model to calculate the ratio of the nonsynonymous substitution rate (dN) to the synonymous substitution rate (dS) (ω = dN/dS) by running CodeML in PAML 4.8a package (Yang, 2007). We also recoded premature stop codons as missing data. Akaike information criterion (AIC) scores were used to select the most appropriate codon frequency model in CodeML. The ACPT gene tree exhibits different topological relationship compared to species tree, which may be unrelated to incomplete lineage sorting. In order to illuminate the detected signal reasonably and accurately, we used a species tree supported by some previous studies (Fig. S1).

Refer to the methods of Springer and Gatesy (Springer & Gatesy, 2018), several different branch categories were considered during selective analyses: (1) One category accounted for “background” branches, which are lineages with intact teeth and an intact copy of ACPT. (2) Nine branch categories to terminal branches with unique inactivating mutations (baleen whales), which lacks teeth. (3) Three branch categories to terminal branches with unique inactivating mutations (pygmy sperm whale, nine-banded armadillo and aardvark), whose enamel has been vestigial. (4) One branch categories were assigned for stem Mysticeti where mineralized teeth were degraded. (5) One branch categories were assigned for crown Mysticeti.

To better understand the selective pressure, a series of evolutionary models were compared in the likelihood. We first use the M0 model (Model A), which assumed that all branches in the phylogenetic tree has a common value, and compare it with the null hypothesis (Model B), which assumed that the common value in the phylogenetic tree is 1. To further understand whether the selective pressure on the lineages leading to pseudogenes was relaxed, we constructed Model C, which assumed that the branches with pseudogene had their own selection pressure ω2, while the background branches without pseudogenization was ω1, and then compared Model C with Model A. To further confirm whether the selective pressure on the lineages leading to pseudogenes was completely relaxed, we build the Model D, which assumed that the branches with pseudogene had their own selection pressure ω2 = 1, while the selective pressure of background branches was ω1, and then compared Model C with Model D.

Estimation of inactivation times

To estimate when ACPT was inactivated in different lineages of Placentalia, the method described in Chou et al. (2002) and Zhang et al. (2010) was used. Among the branches along which the gene became pseudogenes, this method presumes that gene evolves under a selective pressure similar to that in other species until it is inactivated. Next, this gene was presumed to accumulate both nonsynonymous and synonymous mutations at an equal rate. The Ka/Ks (K) value was assessed for this entire branch. The average Ka/Ks value was just for a part of the branch, where the gene was under selection (Ks). In addition, the Ka/Ks value for the rest of part of the branch where the gene evolved neutrally (Kn = 1). Thus, the evolutionary time was weighted by the proportion, for which the gene was evolving under selection (Ts/T) and neutrally (Tn/T): K=Ks×Ts/T+Kn×Tn/T

where T is the time since the split from the last common ancestor (LCA). By selecting the lower and upper bound of the confidence interval for the species divergence time T, which was obtained from TimeTree website (http://www.timetree.org/) to estimate a lower and upper bound for Tn as: Tn=T×(K−Ks)/(1−Ks)

which provides an estimate of how long the ACPT gene has been evolving neutrally.

Results

Characterization of ACPT sequence

A total of 120 sequences were obtained in this study. Due to the poor quality and low coverage of sequences among three pangolins (Manis javanica, M. javanica, Phataginus tricuspis) and one sloth (Choloepus hoffmanni), they were not used for subsequent analysis. However, some inactivating mutations (most of them are indels) were found in these sequences (Fig. S2). The complete protein-coding sequence of ACPT in 116 taxa were used for alignment by PRANK. Interestingly, one or more inactivating mutations (frame-shift mutation, initial codon mutation, premature stop codons, splice site mutations, etc.) were detected in another placental taxa without teeth or without enamel (Fig. 1; Table S4; Fig. S3). For example, among toothless baleen whales, the initial codon mutation (n. ATG → GTG, p. M → V) was found in Balaenoptera borealis, B. physalus, B. musculus, Eschrichtius robustus, Eubalaena glacialis. Meanwhile, premature stop codons were found in B. acutorostrata and B. bonaerensis, frameshift indels were also found in baleen whales. Interestingly, two shared single-base site deletion was found on exon 4 and 5 of ACPT among all living baleen whales (Fig. 1; Fig. S3). The splice site mutations were detected in B. acutorostrata, Eubalaena japonica and Megaptera novaeangliae (Table S4). Whilst, the premature stop codons were found in enamel-less D. novemcinctus and Orycteropus afer. Besides, frameshift indels were found in enamel-less Kogia breviceps.

Figure 1 The inactivating mutation of ACPT gene in toothless/enamel-less mammals.

ICM, initiation codon mutation; Del, deletion; Ins, insertion; PSC, premature stop codon. Images by: Chris huh (http://phylopic.org/image/374accb5-16d5-4cb9-a67a-e881ddfec114/, http://phylopic.org/image/8b73f54f-15e8-41b8-8c9c-46c86a185104/, http://phylopic.org/image/5bfb840e-071f-4a1a-b101-0a747a5453e7/), Creative Commons Attribution ShareAlike 3.0 Unported license; Phylopic (http://phylopic.org/image/3c526436-d40d-45b2-8b43-4f9838a43622/), Public Domain Dedication 1.0 license; Steven Traver (http://phylopic.org/image/5d59b5ce-c1dd-40f6-b295-8d2629b9775e/), Public Domain Dedication 1.0 license.

Except for the species mentioned above, ACPT gene in other species whose teeth are intact were found to be activated. Nevertheless, some crucial amino acids mutation was found in toothed species, such as site 76 has been mutated (R76C) in Neophocaena asiaeorientalis.

Reconstruction of ACPT gene tree

We recovered the ACPT gene tree with well-supported values by using Mrbayes method (Fig. 2). In this gene tree, most of orders have been well reconstructed, and have high support rate, for example, Cetartiodactyla, Perissodactyla, Eulipotyphla, Carnivora, Chiroptera etc. In addition, phylogenetic relationships of higher levels have also been well reconstructed, such as Laurasiatheria, Euarchontoglires, Boreoeutheria and Afrotheria. In this gene tree, bayesian posterior probability (PP) values of nearly 70% nodes are generally greater than 0.70. However, the relationship between some order level were relatively chaotic, such as Lagomorpha didn’t cluster with Rodentia, but as the sister group of Primate; Chiroptera and Carnivora clustered together first, and then they became sister group of Perissodactyla.

Figure 2 The BI phylogenetic relationship of mammalian ACPT gene used in this study.

Nucleotide optimal substitution model: GTR+GAMMA; green box indicates toothless taxa, red boxes indicate enamel-less taxa. Images by: Chris huh (http://phylopic.org/image/374accb5-16d5-4cb9-a67a-e881ddfec114/, http://phylopic.org/image/8b73f54f-15e8-41b8-8c9c-46c86a185104/, http://phylopic.org/image/5bfb840e-071f-4a1a-b101-0a747a5453e7/), Creative Commons Attribution ShareAlike 3.0 Unported license; Phylopic (http://phylopic.org/image/3c526436-d40d-45b2-8b43-4f9838a43622/), Public Domain Dedication 1.0 license; Steven Traver (http://phylopic.org/image/5d59b5ce-c1dd-40f6-b295-8d2629b9775e/), Public Domain Dedication 1.0 license.

Evolutionary analyses among toothless and enamel-less mammals

We carried out the PAML analysis to detect the selective pressure of toothless/enamel-less lineages, and found the selective pressure of these toothless/enamel-less lineages (including ancestral nodes, terminal branches and even the whole toothless/enamel-less group) was significantly higher than that of background branches. For example, the terminal branch of B. physalus: ω1 = 0.116, ω2 = 1.883; the terminal branch of M. novaeangliae: ω1 = 0.116, ω2 = 0.641; the terminal branch of E. robustus: ω1 = 0.116, ω2 = 2.688; the terminal branch of E. glacialis: ω1 = 0.116, ω2 = 0.503. A similar tendency was found in the terminal branches of other baleen whales, and further model comparison shows that the selective pressure of these branches had been completely relaxed. Whilst, much higher selective pressure was detected in the ancestral branch of stem mysticeti (ω1 = 0.120, ω2 = 0.436), even the clade of crown mysticeti (ω1 = 0.116, ω2 = 0.522). Meanwhile, higher selective pressure was detected among enamel-less lineages, such as the terminal branch of D. novemcinctus (ω1 = 0.116, ω2 = 0.206), the terminal branch of O. afer (ω1 = 0.116, ω2 = 0.414), and the terminal branch of K. breviceps (ω1 = 0.116, ω2 = 0.581). And the selective pressure of these branches had been completely relaxed, except for the terminal branch of K. breviceps (Table S5).

ACPT inactivation dates

Estimates of inactivation times for ACPT based on dN/dS ratios and equations in Sharma et al. (2018). The mean estimate for the inactivating time of ACPT on the branch of K. breviceps, D. novemcinctus and O. afer is 12.20–15.52 Ma, 10.18–11.30 Ma and 23.60–28.32 Ma, respectively (Fig. 3). The mean estimate for the inactivation of ACPT on the Mysticeti clade is 14.05–16.30 Ma.

Figure 3 Estimated inactivation times of ACPT vs ENAM.

(A) Dasypus novemcinctus (nine-banded armadillo), (B) Orycteropus afer (aardvark), (C) Kogia breviceps (pygmy spermwhale). The inactivation times of ENAM is from (Meredith et al., 2009; Springer et al., 2019). Images by: Steven Traver (http://phylopic.org/image/5d59b5ce-c1dd-40f6-b295-8d2629b9775e/, http://phylopic.org/image/5d59b5ce-c1dd-40f6-b295-8d2629b9775e/), Public Domain Dedication 1.0 license; Tracy A. Heath (http://phylopic.org/image/6c9cb19d-1d8a-4215-88ba-d49cd4917a5e/), Public Domain Dedication 1.0 license; Jiro Wada (http://phylopic.org/image/30d16233-028d-4a70-9002-db349d73c0bc/), Public Domain Dedication 1.0 license; Pearson Scott Foresman (http://phylopic.org/image/cfee2dca-3767-46b8-8d03-bd8f46e79e9e/), Public Domain Mark 1.0 license; Mo Hassan (http://phylopic.org/image/f266f85c-9c03-4620-8921-2bec8099353a/), Creative Commons Attribution-NonCommercial-ShareAlike 3.0 Unported license.

Discussion

ACPT is a novel candidate gene for studying mammalian tooth loss and enamel loss

The well-conserved gene structure in extant species indicates that this organization and arrangement might be present in the last common mammalian ancestor, which represented the vital function for organisms (Madsen, 2009). In our study, the number of ACPT exons are 11 in placental mammals, which encode 427 amino acids (human ACPT sequence as the reference sequence). Our study collected that four residues (191N, 269N, 330N and 339N) of the extracellular region were for glycosylation, two residues (41H and 289D) directly involved in catalysis (from the UniProt database). In addition, mutation in seven residues were reported that were responsible for AI (Seymen et al., 2016; Smith et al., 2017a) (Fig. S4). Besides, there are three disulfide bond regions, namely, site 159–378, site 214–312, site 353–357. In fact, we detected not only teratogenic mutations but also inactivated mutations in these functional sites and domains. For example, enamel in finless porpoise were degenerated (Ishiyama, 1987), mutation in site 76 (R → C) was found in N. asiaeorientalis. Previous research has confirmed that site 76 mutated into Cys (C) in human ACPT would lead to hypoplastic AI (Seymen et al., 2016), from which this result further supported that teeth in finless porpoise were degenerated in molecular level. Of cause, most obvious characteristics of ACPT is that different types of inactivating mutations were found in toothless and enamel-less mammals, for example, baleen whales, pangolins, sloths and so on (Figs. S2 and S3). Therefore, ACPT could be a candidate gene for AI and studying mammalian tooth loss and enamel loss.

Degeneration or loss of mineralized teeth in LCA of Mysticeti

Fossil evidence shows that the earliest ancestors of baleen whales possessed complete dentitions without baleen (such as Janjucetus and Mammalodon), and then evolved the baleen with teeth (such as Aetiocetus), until the lineages only baleen existed (e.g., Eomysticetus and Micromysticetus) (Fitzgerald, 2006, 2010; Meredith et al., 2011). However, the fact is all living baleen whales lack teeth and instead baleen (Uhen, 2010). This implied that that mineralized teeth were lost or degenerated gradually in the common ancestors of all modern baleen whales (Boessenecker & Fordyce, 2015). In addition, the successive steps of vestigial tooth development was found in the fetal period of living baleen whales (Davit-Béal, Tucker & Sire, 2009; Thewissen, 2018), which was also confirmed by genetic evidence. Molecular sequences of some specific genes, such as AMBN, ENAM, AMELX, AMTN, C4orf26 and ODAM, contain different types of inactivating mutations (e.g., stop codons, frameshift mutations, splice site mutations, etc.) in various mysticete species (Deméré et al., 2008; Meredith et al., 2009; Alhashimi et al., 2010; Gasse, Silvent & Sire, 2012; Meredith, Gatesy & Springer, 2013; Delsuc, Gasse & Sire, 2015; Springer et al., 2019), which is consistent with loss-of-teeth in this group. But none of the inactivating mutations are shared by all living mysticetes species. Meredith et al. (2011) found a common insertion of CHR-2 SINE retroposon in MMP20 gene among all living baleen whales. Previous study has been confirmed that mutations or deletions of MMP20 gene would result in thin and brittle enamel layer (Caterina et al., 2002). Based on this result, they confirmed the hypothesis that mineralized teeth were lost or degenerated in the common ancestor of crown Mysticeti in the molecular level.

In this research, we also identified different inactivating mutations was detected among all mysticete species in ACPT gene, among which two shared single-base sites deletion were found on exon 4 and 5 of ACPT among all living baleen whales, which result in loss of function. Some studies have confirmed that ACPT gene is responsible for the development of enamel, and mutations can also lead to amelogenesis imperfecta (Choi et al., 2016; Seymen et al., 2016; Smith et al., 2017a). Similar to the result of Meredith et al. (2011), our study supported the hypothesis that mineralized teeth were lost or degenerated in the common ancestor of all extant baleen whales.

Is inactivation of ACPT neutral or adaptive?

The degeneration and/or loss of some morphological structures (such as limbs, teeth, and eyes, etc.) is a complex process that may result from the relaxation of the negative selection (neutral evolution), adaptive evolution (direct natural/positive selection to conserve energy and/or eliminate the disadvantageous effects of morphological structure), and/or gene pleiotropy (indirect selection on another traits) (Wang, Grus & Zhang, 2006; Zhang, 2008; Krishnan & Rohner, 2017). In some conditions, evolutionary change also results from differences in the reproductive success of individuals with different genotypes (Olson, 1999). Sharma et al. (2018) revealed that evolutionary gene losses are not only a consequence, but may also be causally involved in phenotypic adaptations. By estimating the inactivation time of pseudogenes, and comparing with oldest fossil records, we might be able to speculate whether gene inactivation is due to the adaptive or neutral selection after the loss of phenotype.

The record of enamel-degenerated armadillo fossil is significantly earlier than the estimated time of ACPT inactivation (10.18–11.30 Ma) (Ciancio, Vieytes & Carlini, 2014), which suggested gene loss as a consequence of adaptation is likely the result of the relaxation of the negative selection. The results further supported the previous study (Sharma et al., 2018). Besides, during the tooth evolution, some enamel-related genes (e.g., ODAM, ENAM, AMBN) also have gone through the similar evolutionary trajectory. By integrating different results from different methods, we may better understand the evolution of teeth and enamel. The inactivation time of ENAM (~45.5 Ma) and ODAM (~40.43 Ma, range 36.38–45.45 Ma) is much earlier than inactivation date for ACPT in armadillo (Springer et al., 2019). ACPT inactivation is later than the fossil record, conversely, the inactivation time of ENAM is relatively earlier than the fossil record, which implied the various mechanisms of enamel loss in armadillo. Here, the inactivation of ENAM gene might be the causes of degeneration/loss of tooth enamel in armadillos, ACPT inactivation might be the consequence of enamel loss.

For O. afer, even the inactivation date for ACPT (23.60–28.32 Ma) is relatively younger than inactivation dates for ENAM (28.8–35.3 Ma) and ODAM (~30.7 Ma) in O. afer (Meredith et al., 2009; Springer et al., 2019). However, the estimated inactivation times by ACPT, ODAM and ENAM gene markers are all earlier than the oldest fossil record of aardvark (O. minutus, ~19 Ma) (Patterson, 1975). It should be suggested that gene loss may be the reason, not the consequence, for degeneration and/or loss of enamel. Moreover, due to the difference of species number, sequences quality and topological structure of species tree, the result of ACPT inactivation time is different from the result of Sharma et al. (2018).

Cetacean includes both toothless Mysticeti and enamel-less Kogia. Relaxation of selective pressure was detected in both crown and stem Mysticeti (Table S5), which is consistent with the archaic toothless mysticete, namely, all stem Mysticeti were toothless. For example, Eomysticetus whitmorei, an edentulous species, was the geologically oldest mysticete (Deméré et al., 2008). Molecular evidence shows ACPT has been lost its function in LCA of Mysticeti. However, the inactivation time of ACPT in Mysticeti is 14.05–16.30 Ma, which is much younger than the toothless mysticete (~30 Ma) and the split of Mysticeti (~25.9 Ma). Obviously, this is not consistent with the facts. It might be associated with relatively lower rates of frameshift accumulation during evolution of mysticete pseudogenes and long lifespan of mysticete (Meredith et al., 2009, 2011). Whether adaptive or neutral, the shared single-base site deletion in ACPT fills an important gap in our understanding of the macroevolutionary transition leading from the LCA of crown Cetacean to the LCA of crown Mysticeti. Stem physeteroids (sperm whales) are known from the Miocene and had teeth with enamel (Bianucci & Landini, 2010). Our results provide support for loss of the intact ACPT in K. breviceps. ACPT was reported that play key roles in amelogenesis and differentiation of odontoblasts (Choi et al., 2016; Seymen et al., 2016; Smith et al., 2017a). Our result is in line with the enamel-less morphological structure in K. breviceps.

Conclusions

We detected the different types of inactivated mutation in ACPT. Furthermore, selective pressure uncovered that the selective constraints have been relaxed among all toothless and enamel-less lineages. In addition, our results supported the hypothesis that mineralized teeth were lost or degenerated in the common ancestor of crown Mysticeti through two shared single-base sites deletion in exon 4 and 5 of ACPT among all living baleen whales. Together with our evidence, ACPT might be a good marker to research the mechanism of tooth loss. By comparing the molecular time with the fossil time, we found there might be different mechanisms of degeneration of tooth/among toothless and enamel-less lineages during evolution, which is needed further researches.

Supplemental Information

Supplemental Information 1 Supplemental Tables.

Click here for additional data file.

Supplemental Information 2 Supplementary Figures.

Click here for additional data file.

We thank members of the Jiangsu Key Laboratory for Biodiversity and Biotechnology, Nanjing Normal University, for their contributions to this paper. The authors thank Mr. Xinrong Xu, Dr. Di Sun and Dr. Ran Tian, Dr. Zepeng Zhang, Dr. Simin Chai and Dr. Zhenpeng Yu for some helpful discussion. Special thanks to Dr. Zhengfei Wang for technical supports.

Additional Information and Declarations

Competing Interests

Author Contributions

Data Availability

The authors declare that they have no competing interests.

Yuan Mu conceived and designed the experiments, performed the experiments, prepared figures and/or tables, authored or reviewed drafts of the paper, and approved the final draft.

Xin Huang performed the experiments, prepared figures and/or tables, and approved the final draft.

Rui Liu performed the experiments, prepared figures and/or tables, and approved the final draft.

Yulin Gai analyzed the data, authored or reviewed drafts of the paper, and approved the final draft.

Na Liang analyzed the data, prepared figures and/or tables, and approved the final draft.

Daiqing Yin analyzed the data, prepared figures and/or tables, and approved the final draft.

Lei Shan analyzed the data, authored or reviewed drafts of the paper, and approved the final draft.

Shixia Xu conceived and designed the experiments, authored or reviewed drafts of the paper, and approved the final draft.

Guang Yang conceived and designed the experiments, authored or reviewed drafts of the paper, and approved the final draft.

The following information was supplied regarding data availability:

This work used publicly available data. Sequences are available at the NCBI database (https://www.ncbi.nlm.nih.gov/), ENSEMBL (http://www.ensembl.org/index.html?redirect=no), and OrthoMaM v10b (http://orthomam1.mbb.univ-montp2.fr:8080/OrthoMaM_v10b5/). Accession numbers are available in the Supplemental Files.

The topological species tree is based on the OrthoMaM database and previous studies (Celine et al., 2019; Waddell, Kishino & Ota, 2001; Sergey et al., 2007; Zhou et al., 2012; Gatesy et al., 2013; Kuntner, May-Collado & Agnarsson, 2011).

Data that we generated during analysis appear in the Results section and in the Supplemental Files.

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
