# Peer review of "ACPT gene is inactivated in mammalian lineages that lack enamel or teeth"

_PeerJ, doi:10.7717/peerj.10219_

## Round 0.1 · original submission · Major Revisions

Dear Dr. Mu and colleagues:

Thanks for submitting your manuscript to PeerJ. I have now received two independent reviews of your work, and as you will see, the reviewers raised some concerns about the research. Despite this, these reviewers are optimistic about your work and the potential impact it will have on research studying dentition in mammals. Thus, I encourage you to revise your manuscript, accordingly, taking into account all of the concerns raised by both reviewers.

Please minimize over-speculation, particularly in light of an ambiguous fossil record.

It appears that key organisms are missing in your comparative analyses (pangolin was brought up by both reviewers). Please add sequences from as many enamel-less and toothless mammals as possible (especially those for which sequenced genomes are available, like pangolin, sloth, and others). Your decisions for excluding (and overall taxon sampling) need to be justified, and analyses should be robust such that perturbations of your analyzed taxa do not drastically change your results and interpretations.

Reviewer 2, in particular, has some expert advice on many aspects of your analyses. Please address these suggestions accordingly.

Importantly, please ensure that an English expert has edited your revised manuscript for content and clarity. Please also ensure that your figures and tables contain all of the information that is necessary to support your findings and observations. Please make sure that all relevant literature is cited, and that your findings are compared to what is already established in the literature.

Therefore, I am recommending that you revise your manuscript, accordingly, taking into account all of the issues raised by the reviewers.

I look forward to seeing your revision, and thanks again for submitting your work to PeerJ.

Good luck with your revision,

-joe

Reviewer 1 ·

Basic reporting

Mu et al. provide a potentially interesting study on the evolutionary loss of the ACPT gene in enamel-less and edentate mammals.

Overall, I like the paper but there is much work that still needs to be done. The writing style needs work as does the English. Switching of tenses is also prevalent throughout paper - some examples are given below. Figures are fine but where did the images come from?

Experimental design

The question is well defined but more investigation is required.

What about Manis spp. (pangolins)? They are edentulous as well. There is no discussion on this group at all. Why are there no sequences from this species in this study given that two genomes have been sequenced?

Validity of the findings

I question, whether or not there is anything known about different isoforms of ACPT. This obviously could directly impact results of this study if there is evidence of an isoform that excludes exon 4 and 5 or any other exons that have deleterious mutations. If there is nothing known than that should be stated as well.

I find the whole section "Is inactivation of ACTP neutral or adaptive" to be very speculative. Dating methods between the different studies used both in this paper and others are not the same. This might also be influencing the discussion in this section - the assumptions of the equations will strongly affect the resulting date estimates. This section is also assuming that the oldest describe fossils are indeed the oldest representative of that lineage which we know is probably not the case given the incompleteness of the fossil record.

Additional comments

Italicize Amelogenesis imperfecta throughout paper

line 48: etc. not needed
line 49-50: Awkwardly written
line 77: Not necessarily true. ACTP could play an important role in amelogenesis but simply being pseudogenized does not guarantee that

line 78: delete "To our knowledge"
line 79: delete "where instead of" replace with "and instead have"

line 111: Alignment and phylogenetic analysis of mammalian ACPT
No mention of checking for convergence among chains in Bayesian analysis. Why a 50% majority rule tree?

line 129: do you mean "related" not "unrelated"

line 151: Estimation of inactivation times
This section is not properly cited. The method Sharma et al used was published by two other studies - see the paper for details.

line 219: I'm not following this sentence

line 222: " Our study highlighted that four
223 residues (191N, 269N, 330N and 339N) of the extracellular region were for glycosylation, two 224 residues (41H and 289D) directly involved in catalysis. "
Where were the methods for this result or what paper did this come from??

line 238: replaced "showed" with "shows" - this paper

line 304: replace "Furtherly" with "Furthermore"

Italicize species names throughout

Italicize gene names throughout

Reviewer 2 ·

Basic reporting

Generally this is fine. The authors should get help to fix English grammar, vocabulary usage, conjugation of verbs, nouns as adjectives, etc. throughout the manuscript. The text is not so far off but does need extensive work in this context and is pretty rough in parts in terms of grammar. I generally do not make these sorts of changes in manuscript ad hoc review.

Experimental design

The experimental design is generally fine in terms of methods, but additional sequences should be included from various enamel-less and toothless mammals that have sequenced genomes (pangolin, sloth, and perhaps others).

Validity of the findings

1) The authors should seek to clearly note what Sharma et al. (2018) noted about inactivation of ACPT in mammals with no teeth or enamel and distinguish this clearly from what this new study shows relative to that earlier study that originally documented the gene knockouts in mammals without teeth or enamel. For example, extending analyses to more mysticete whale species and to pygmy sperm whale, etc. and inference of inactivation times. Did Sharma et al. (2018) estimate inactivation times as well? If so, how do dates and error bars on dates differ?
2) line 33. " which is earlier than the oldest pangolin" should be " which is earlier than the oldest aardvark".
3) lines 33-36. It is not clear in the abstract what is meant by " the oldest fossil time". Perhaps this is a word choice issue and is ambiguous as written in the context of these sentences. i.e., "oldest fossil time" of what?
4) Is it known exactly what the ACPT gene does in enamel formation? Or, is this still not well studied by experimental means in past literature? Are there hypotheses regarding what role this gene plays in tooth development?
5) Why was only one species of Xenarthran analyzed in this paper? Why are there no pangolin sequences in the study? These are critical excluded species that are toothless (pangolins, anteaters) or lack enamel (sloths and other aramadillos). Are genomes for these additional species not available? Or, is the ACPT gene completely lacking from currently available genome sequences? If the latter, are flanking genes relative to ACPT found in these toothless or enamel-less mammals? My understanding is that Dasypus is an armadillo with very recent degradation of enamel based on previous morphological, paleontological, and molecular work and that multiple losses of enamel occurred within the armadillo clade. Are the authors up to date on this literature?
6) line 263. The subtitle of this section is " Is inactivation of ACPT neutral or adaptive?" but I think this challenging question was not really answered adequately in this section. I did not find the authors' arguments here to be compelling. There are multiple hypothesized losses of enamel in Xenarthra, not just one in the common ancestor of this clade. Because the authors included just one Xenarthran in their analysis and this is in an armadillo that seems to have lost enamel very recently, the text of the authors does not make sense to me. I think much more work has to be done here to make a compelling argument (i.e., more sequences from xenarthrans need to be included from sloths, anteaters, and more armadillo genera). I did not follow the logic of the statement, " However, estimates for ACPT, ODAM and ENAM inactivation are both older than the oldest fossil aardvark, O. minutus, which is ~19Ma (Patterson, 1975). It strongly suggested that gene loss may be the reason, not the consequence, for degeneration and / or loss of enamel, which is different from the result of Sharma et al. (Sharma et al., 2018)." What did Sharma et al. say and how does this differ logically from the interpretation of the authors here. I am also not understanding how the very young date for inactivation of ACPT was calculated. What was the time range used for the ancestral branch to Mysticeti, and what was this based on? Isn't the basal node of Mysticeti estimated from previous work to be older than 15 MY, which is the authors' estimate for inactivation on the stem lineage of Mysticeti? Or, maybe I am missing something here. At any rate, this should be clarified in revision. Also, there is recent evidence that the "toothless" mysticete Eomysticetus is in a clade (Eomysticetidae) in which some close relatives have vestigial teeth at the tips of their jaws. This region of anatomy is not, I think, well preserved enough in Eomysticetus to tell whether this genus was truly toothless?
7) line 266. The author's definition here seems off. " adaptive evolution (direct natural / positive selection to conserve energy and / eliminate the disadvantageous effects of morphological structure)" What about reproductive fitness implied by different genotypes? Their definition is unconventional even if it is as in the references given.

Additional comments

Yuan Mu et al. analyze sequence data for the ACPT gene in a phylogenetic context to characterize inferred gene inactivations over the history of Mammalia. The authors note that they seek to analyze the "mechanisms" of gene inactivation as related to tooth loss in the abstract and in the introduction. This is a challenge as it is difficult to assess cause and effect when so many genes have been inactivated on the same evolutionary branches where inferred losses of enamel or teeth have occurred in mammalian phylogeny. The authors document knockouts of ACPT in baleen whales, pygmy right whale, aardvark, and armadillo which are all characterized by lack of enamel or degraded enamel. This work follows a previous paper that first noted some of these inactivations (Sharma et al., 2018) that was recently published using broad genomic screens. So, as the authors note, it was previously known that inactivations of ACPT in mammals without enamel were present, but the authors provide extensive further detail on these events within the context of phylogenetic trees from many more mammalian species (116 species) that were extracted from published genomes and estimate inactivation times using dN/dS ratios on branches where loss of enamel is inferred. The paper is pretty simple and straightforward and represents a contribution to the field, but the new insights presented are perhaps not great given previous work on this gene by Sharma et al. (2018). An important issue is point #6 below. Why were pangolin and anteater sequences not included in this study, as well as sloth sequences? These are critical species without teeth or enamel that have published genomes? In general the authors methods seem sound, and the results and discussion are simply stated and not overinterpreted.

1) The authors should seek to clearly note what Sharma et al. (2018) noted about inactivation of ACPT in mammals with no teeth or enamel and distinguish this clearly from what this new study shows relative to that earlier study that originally documented the gene knockouts in mammals without teeth or enamel. For example, extending analyses to more mysticete whale species and to pygmy sperm whale, etc. and inference of inactivation times. Did Sharma et al. (2018) estimate inactivation times as well? If so, how do dates and error bars on dates differ?
2) The authors should get help to fix English grammar, vocabulary usage, conjugation of verbs, nouns as adjectives, etc. throughout the manuscript. The text is not so far off but does need extensive work in this context and is pretty rough in parts in terms of grammar. I generally do not make these sorts of changes in manuscript ad hoc review.
3) line 33. " which is earlier than the oldest pangolin" should be " which is earlier than the oldest aardvark".
4) lines 33-36. It is not clear in the abstract what is meant by " the oldest fossil time". Perhaps this is a word choice issue and is ambiguous as written in the context of these sentences. i.e., "oldest fossil time" of what?
5) Is it known exactly what the ACPT gene does in enamel formation? Or, is this still not well studied by experimental means in past literature? Are there hypotheses regarding what role this gene plays in tooth development?
6) Why was only one species of Xenarthran analyzed in this paper? Why are there no pangolin sequences in the study? These are critical excluded species that are toothless (pangolins, anteaters) or lack enamel (sloths and other aramadillos). Are genomes for these additional species not available? Or, is the ACPT gene completely lacking from currently available genome sequences? If the latter, are flanking genes relative to ACPT found in these toothless or enamel-less mammals? My understanding is that Dasypus is an armadillo with very recent degradation of enamel based on previous morphological, paleontological, and molecular work and that multiple losses of enamel occurred within the armadillo clade. Are the authors up to date on this literature?
7) line 263. The subtitle of this section is " Is inactivation of ACPT neutral or adaptive?" but I think this challenging question was not really answered adequately in this section. I did not find the authors' arguments here to be compelling. There are multiple hypothesized losses of enamel in Xenarthra, not just one in the common ancestor of this clade. Because the authors included just one Xenarthran in their analysis and this is in an armadillo that seems to have lost enamel very recently, the text of the authors does not make sense to me. I think much more work has to be done here to make a compelling argument (i.e., more sequences from xenarthrans need to be included from sloths, anteaters, and more armadillo genera). I did not follow the logic of the statement, " However, estimates for ACPT, ODAM and ENAM inactivation are both older than the oldest fossil aardvark, O. minutus, which is ~19Ma (Patterson, 1975). It strongly suggested that gene loss may be the reason, not the consequence, for degeneration and / or loss of enamel, which is different from the result of Sharma et al. (Sharma et al., 2018)." What did Sharma et al. say and how does this differ logically from the interpretation of the authors here. I am also not understanding how the very young date for inactivation of ACPT was calculated. What was the time range used for the ancestral branch to Mysticeti, and what was this based on? Isn't the basal node of Mysticeti estimated from previous work to be older than 15 MY, which is the authors' estimate for inactivation on the stem lineage of Mysticeti? Or, maybe I am missing something here. At any rate, this should be clarified in revision. Also, there is recent evidence that the "toothless" mysticete Eomysticetus is in a clade (Eomysticetidae) in which some close relatives have vestigial teeth at the tips of their jaws. This region of anatomy is not, I think, well preserved enough in Eomysticetus to tell whether this genus was truly toothless?
8) line 266. The author's definition here seems off. " adaptive evolution (direct natural / positive selection to conserve energy and / eliminate the disadvantageous effects of morphological structure)" What about reproductive fitness implied by different genotypes? Their definition is unconventional even if it is as in the references given.

---

## Round 0.2 · Minor Revisions

Dear Dr. Mu and colleagues:

Thanks for revising your manuscript. The reviewer is very satisfied with your revision (as am I). Great! However, there is a minor issue to consider. Please address this ASAP so we may move towards acceptance of your work.

Best,

-joe

Reviewer 2 ·

Basic reporting

This is fine. English grammar and word usage could still use some help.

Experimental design

no comment

Validity of the findings

This is generally fine. I think there is still some confusion on stem baleen whales (Mysticeti). Some have teeth, and even eomysticetids that have been described in the past as toothless show some evidence of having teeth at the tips of their jaws in at least one species (see Boessenecker papers from 2015). This affects some of the discussion section on whales, and it would be best to get this right.

Additional comments

I think the authors have generally dealt with the comments from my initial review in an adequate way.

---

## Round 0.3 · accepted · Accept

Dear Dr. Mu and colleagues:

Thanks for revising your manuscript based on the concerns raised by the reviewers. I now believe that your manuscript is suitable for publication. Congratulations! I look forward to seeing this work in print, and I anticipate it being an important resource for groups studying dentition in mammals. Thanks again for choosing PeerJ to publish such important work.

Best,

-joe